# Efficient GHz electro-optical modulation with a nonlocal lithium niobate metasurface in the linear and nonlinear regime

Agostino Di Francescantonio [1], Alessandra Sabatti [2], Helena Weigand[2], Elise Bailly-Rioufreyt [2], Maria Antonietta Vincenti [3], Luca Carletti [3], Jost Kellner [2], Attilio Zilli [1], Marco Finazzi [1], Michele Celebrano [1] ✉ & Rachel Grange [2] ✉

Electro-optical modulation is essential in optical signal processing and laser technology, yet modulators based on the Pockels effect in flat optics lag behind bulk and integrated platforms in efficiency and speed. We bridge this gap realizing a metasurface based on lithium niobate ($LiNbO_3$) on insulator that leverages on resonances with quality-factor as high as 8000 to achieve fast electrical modulation of both linear and nonlinear optical properties. $LiNbO_3$, well known for its high nonlinear susceptibility and wide transparency window across the infrared and visible spectrum, is employed to realize an asymmetric, one-dimensional array of nanowires, exhibiting resonances with linewidth <0.2 nm. The metasurface achieves a reflectivity modulation around 0.1, with a modulation efficiency, defined as relative modulation *per* applied Volt, larger than 0.01 $V^{-1}$ on a −3 dB (−6 dB) bandwidth of about 800 MHz (1.4 GHz). Additionally, we demonstrate more than one order of magnitude intensity modulation of the second harmonic seeded by a continuous-wave laser, with a modulation efficiency of about 0.12 $V^{-1}$. This dual modulation capability, rooted in the interplay between optical resonances and electric field manipulation, holds significant potential for cutting-edge applications in high-speed photonics, nonlinear optics, and reconfigurable communication systems.

Optical metasurfaces represent a promising platform for scaling optical components down to the micrometer thickness[1]. While most of the demonstrated optical operations rely on passive metasurfaces, whose functionalities are not tunable after fabrication, applications like beam steering, LiDAR, and free-space optical communication demand active control over various light properties, including amplitude, phase, polarization, and directionality. As a result, there is a growing emphasis on the development of reconfigurable metasurfaces aimed at creating ultra-compact free-space photonic devices capable of dynamically adjusting these essential parameters[2]. Several physical mechanisms have been explored to achieve this goal, including metasurfaces integrated with liquid crystals[3], phase-change materials[4], thermo-optic control[5,6], electro-optic (EO) modulation[7,8], as well as all-optical modulation by means of absorption of femtosecond laser pulses[9,10], and interferometric control[11]. Although several of these approaches are very suggestive, EOM remains the most promising in terms of applications due to its compatibility with CMOS-based platforms. EOM relies on nonlinear effects, like the Pockels (or linear

[1]Politecnico di Milano, Physics Department, Milano, Italy. [2]ETH Zürich, Department of Physics, Institute for Quantum Electronics, Optical Nanomaterial Group, Zürich, Switzerland. [3]Università di Brescia, Department of Information Engineering, Brescia, Italy. ✉e-mail: michele.celebrano@polimi.it; grange@phys.ethz.ch

electro-optic) effect and the Kerr (or quadratic electro-optic) effect. Both effects result in a change of the refractive index $n$, induced by a static or low-frequency electric field. In the case of the Pockels effect, this change is linear with respect to the quasi-static electric field, $\Delta n = -\frac{1}{2} n^3 r_{\text{eff}} E_{\text{EO}}$ [12], where $r_{\text{eff}}$ is the material effective electro-optic coefficient, and $E_{\text{EO}}$ is the amplitude of the driving electric field. A key advantage of the Pockels effect is its ability to achieve modulation speeds more than a hundred of GHz[13,14], with the effective observed modulation bandwidth primarily limited by the electrical design of the device actuation system. This makes the Pockels effect particularly promising for high-speed applications, such as optical communications.

Among the available material platforms, lithium niobate (LiNbO$_3$) in crystalline form is already widely used for bulk photonics devices[15], thanks to its wide transparency window from the mid-infrared to ultraviolet and its large electro-optic coefficient ($r_{33} \simeq 35$ pm/V[16]) and second-order nonlinear coefficient ($d_{33} \simeq -27$ pm/V[17]). Several important applications of LiNbO$_3$ are based on its optical nonlinearity, like EOM or frequency conversion via optical harmonic generation, such as second-harmonic generation (SHG), or spontaneous parametric down conversion. Advances in fabrication techniques have made thin-film LiNbO$_3$ available for integrated[18] and metasurface-based photonic devices[19]. Yet, in contrast to on-chip modulators based on LiNbO$_3$ waveguides, which routinely achieve efficient modulation up to the GHz range[14,20,21], the development of efficient free-space electro-optic modulators based on subwavelength platforms remains an open challenge. This can be primarily ascribed to the inherently short interaction lengths and the modest refractive index changes associated with typical electro-optic materials.

Recent efforts to expand the materials palette for EOM free-space platforms include, for example, barium titanate (BTO), which has been investigated as an electro-optic bottom-up metasurface[22]. However, the potential benefits of its large electro-optic coefficient ($r_{\text{BTO}} = 148$ pm/V) are hindered by non-standard fabrication techniques and by its variability depending on the specific method[23]. Alternative approaches, such as electro-optic polymers[24,25], while exhibiting large coefficients, lack robustness for applications like automotive or under high laser exposure[26]. As a result, LiNbO$_3$ is still the most reliable material for electro-optic modulation, owing to its remarkable non-linear optical properties and established top-down fabrication processes.

Metasurfaces offer a viable route to maximize the effects associated with refractive index modulation[27], thanks to enhanced light−matter interactions. In particular, nonlocal metasurfaces, characterized by spectrally sharp resonances such as lattice resonances[28], guided mode resonances (GMRs)[29], and bound states in the continuum (BICs)[30,31], are extensively investigated as potential free-space modulators. Recently, Benea-Chelmus et al. demonstrated, for the first time, free-space GHz modulation from a Si metasurface supporting BICs and GMRs embedded in an organic, electro-optic layer[8]. However, achieving a modulation efficiency greater than $10^{-2}$ with CMOS-compatible voltages (up to 5 V) remains a non-trivial task. This challenge is further complicated by the requirement to realize high-quality factor ($Q$) structures and to couple them effectively to free-space radiation[30]. In addition to boosting the quality factor of optical resonances, modulation efficiency can also be improved by optimizing the static electric field distribution $E_{\text{EO}}$ by means of specific electrode designs, although this often comes at the expense of a reduced device bandwidth[28,32].

In this study, we leverage high-quality factor resonances in non-local metasurfaces based on monolithic lithium niobate on insulator (LNOI) platform to realize efficient intensity modulation of free-space signals within the telecom C band. Our experimental results reveal a modulation efficiency exceeding 10%, driven by less than 10 V and an operational bandwidth of about 1 GHz. Beyond linear signal

modulation, our system also effectively modulates nonlinear upconverted signals, namely SHG. Notably, we find that the second harmonic signal exhibit significantly larger modulation compared to linear signals, as the nonlinear response originates directly from the material itself. Moreover, the squared dependence of the SHG on the fundamental field enhances the sensitivity to variations in the resonant properties of the system[33]. By harnessing the nonlinear characteristics of lithium niobate, we demonstrate electro-optic modulation of SHG excited by a continuous-wave (CW) pump laser, achieving a striking 110% intensity modulation. This advancement paves the way for the development of a novel class of modulators for upconverted signals, offering exciting possibilities for future photonic technologies.

## Results
### Design and linear characterization
Figure 1 illustrates the design and working principle of our device. We investigate a nonlocal lithium niobate on insulator (LNOI) metasurface made by asymmetric, periodically arranged nanostripes obtained from an x-cut LiNbO$_3$ thin film on a finite SiO$_2$ layer on top of a Si substrate (Fig. 1a). We aim at observing efficient and fast electric modulation of the metasurface optical properties induced by a control voltage, $V_{\text{EO}}$ applied to the metasurface. To this aim, in-plane, gold contacts generate an electric field ($\mathbf{E}_{\text{EO}}$) aligned along the z-axis to exploit the largest component of the electro-optic tensor $r_{33}$ of LiNbO$_3$. To obtain a significant intensity modulation, the structure must sustain resonant modes, which are spectrally shifted when the refractive index is modulated by an applied voltage (Fig. 1b). Furthermore, at a given wavelength $\lambda_0$, a device with large quality factor $Q = \lambda_0/\delta\lambda_0$ features a resonance with a narrow spectral width $\delta\lambda_0$ and, consequently, a large sensitivity to small variations of the refractive index induced by the applied voltage.

To this aim, we exploit high $Q$ resonances hosted by a dielectric waveguide slab − namely, a high refractive index layer (LiNbO$_3$), comprised between two low-index claddings (air and SiO$_2$)−patterned with periodically arranged nanowires (i.e. a diffraction grating)[34]. Figure 2a shows a sketch of the elementary cell of the metasurface with the relevant geometrical parameters. A planar dielectric waveguide is known to host guided modes whose dispersion lies below the light line, fully confining the radiation in the slab volume with no access from

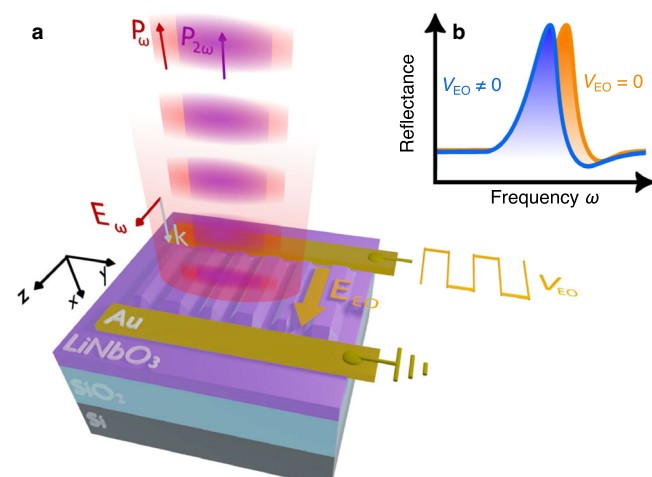

**Fig. 1 | Linear and nonlinear electro-optic modulation with a lithium niobate metasurface. a** Sketch of modulated reflected power of a beam at fundamental frequency $P_\omega$ and of its second harmonic $P_{2\omega}$ from a x-cut LiNbO$_3$ on a SiO$_2$-Si substrate. The modulation is induced by an applied voltage ($V_{\text{EO}}$), which produces a driving electric field $\mathbf{E}_{\text{EO}}$ parallel to the optical field $\mathbf{E}_\omega$. **b** Illustration of a resonance shift induced by electro-optic effect.

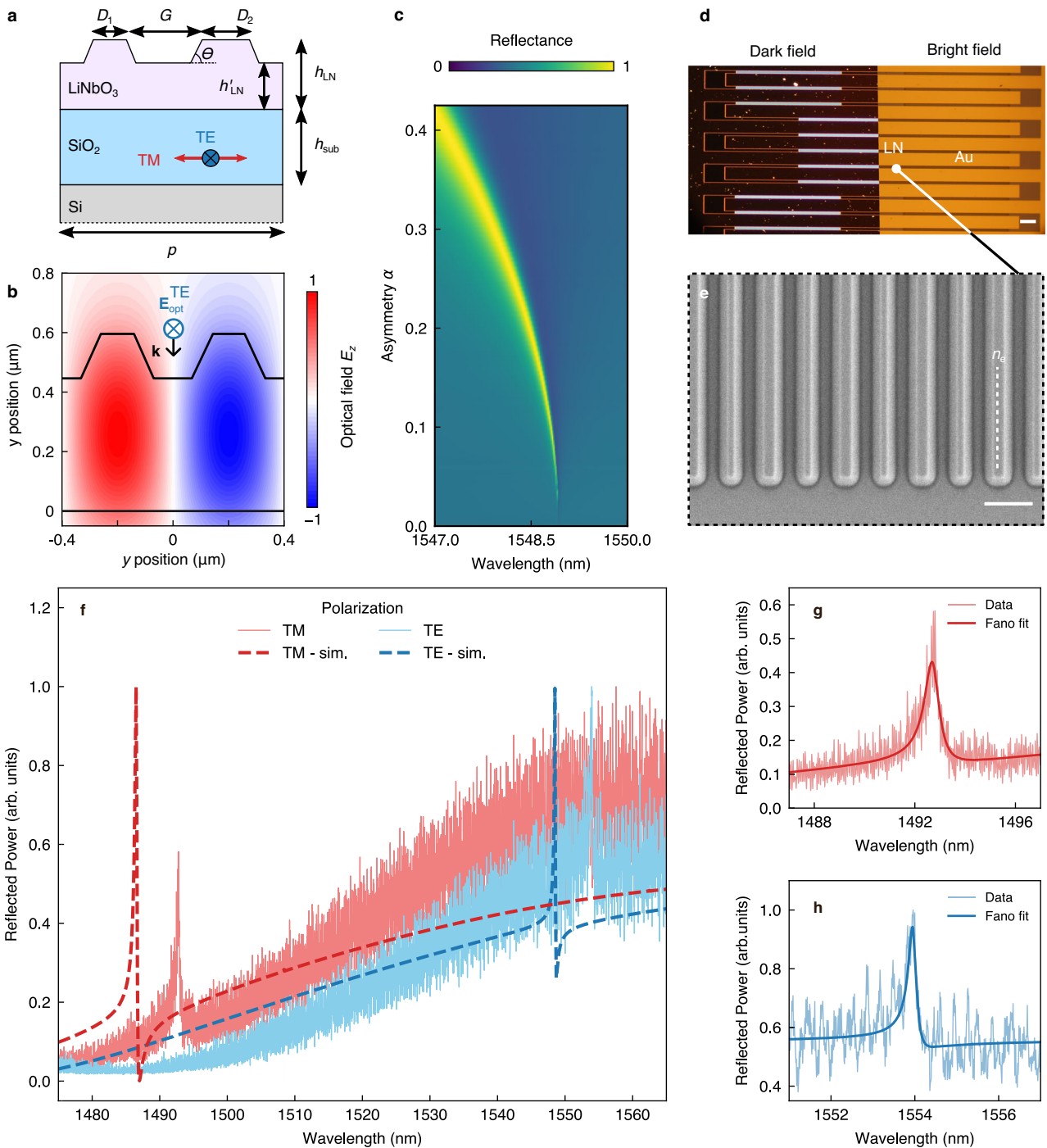

**Fig. 2 | Sample layout and linear optical characterization. a** Schematic cross-section of a single unit cell of the metasurface. All the samples have the same periodicity $p = 800$ nm and LiNbO$_3$ height before ($h_{LN} = 600$ nm) and after patterning ($h'_{LN} = 450$ nm) with a substrate thickness, $h_{sub} = 2$ µm. The angle $\theta = 65°$ accounts for the sidewall slopes introduced by the physical etching. The variable parameters from sample to sample are the filling factor $FF = 1 - (D_1 + D_2)/p)$ and the asymmetry parameter $\alpha = (D_2 - D_1)/(D_2 + D_1)$. Red and blue arrows indicate the polarization of the incident wave that excites the TM (parallel to the LN ordinary axis) and TE (parallel to the LN extraordinary axis) mode, respectively. **b** COMSOL simulation of the optical field $z$-component of the optical field $\mathbf{E}_{opt}$, excited at the quasi-BIC resonance for FF = 0.3 and $\alpha = 0$. Incident optical field $\mathbf{E}_{opt}$ is TE-polarized and propagates with $\mathbf{k} \parallel (-x)$. **c** COMSOL plane-wave simulation of the metasurface reflectance depending on the asymmetry parameter $\alpha$, with fixed FF = 0.3. **d** Dark

field and bright-field images in real colours of the LiNbO$_3$ chip. The LiNbO$_3$ metasurfaces are located in the areas with strong scattering (bright areas in dark field). The white scale bar on the bottom right is equal to 100 µm. The orange lines in the bright field image identify the edges of the interdigitated electrodes. **e** Scanning-electron micrograph showing the metasurface obtained by patterning the LiNbO$_3$ thin film. The scale bar on the bottom right is equal to 500 nm. The white, dashed line indicates the LiNbO$_3$ extraordinary axis. **f** Normalized reflection spectra of the sample with FF = 0.3 and $\alpha = 0.19$. The solid lines show experimental data obtained by sweeping the laser wavelength in steps of 10 p.m., with electric field exciting the TE (blue solid line) and TM (red solid line) mode. The dashed lines are the corresponding numerical simulations. **g, h** Zoom on spectra shown in panel f (thin lines) fitted with Fano profiles (thick lines) for TM and TE polarization, respectively.

external radiation sources. The reduction of translational symmetry introduced by the grating causes a folding of the guided modes dispersion in the first Brillouin zone, ending up above the light line, and consequently evolving in optically accessible guided mode resonances (GMRs)[34,35]. A mirror symmetry of the grating with respect to the $xz$ plane yields a dark fundamental mode at the Γ point (i.e., at normal incidence) due to the odd electric field distribution (see Fig. 2b), namely becoming a symmetry-protected BIC. By introducing a small asymmetry in the nanowires' widths, we further reduce the translational symmetry of the system, turning the BIC into a leaky, quasi-BIC with finite $Q$ (see Fig. 2c), which can now be excited at the Γ point.

The design optimization has been performed initially by means of rigorous coupled-wave analysis, typically applied to solve scattering from periodic dielectric structures[36]. The EO effect and the SHG are then modelled with a finite-element commercial software (COMSOL Multiphysics®, see Methods and Supplementary Information S2). By setting the unit cell periodicity as $p = 800$ nm (see Fig. 2a), the metasurface features the fundamental TE21 resonant mode at the Γ point in the communication C band (i.e, 1530 to 1565 nm). This mode shows up in the reflectance spectrum when the structure asymmetry $\alpha = (D_2 - D_1)/(D_2 + D_1)$ is nonzero, with a linewidth broadening with increasing $\alpha$ (see Fig. 2c)[37]. We set $D_1$ and $D_2$ to have a fill factor $FF = 1 - (D_1 + D_2)/p = 0.3$, corresponding to a fundamental TE mode around 1550 nm, and $\alpha = 0.2$, to realize a theoretical $Q = 11000$. Such a value for the $Q$ factor is selected since it allows more tolerance in terms of fabrication accuracy and better resonance coupling in the experiment. Nevertheless, $Q$ is sufficiently high to achieve efficient EO modulation, since the associated mode linewidth ($\delta\lambda_0 \simeq 0.15$ nm) is comparable with the expected EO shift induced by CMOS-compatible voltages, assuming a tuning sensitivity $\Delta\lambda_{EO}/\Delta V_{EO} \approx 0.01$ nm/V[28,30].

The metasurfaces were realized on a commercial LNOI stack with 600 nm-thick LiNbO₃ film on top of a SiO₂ layer with thickness $h_{sub} = 2$ μm (see Fig. 2d, e) following a top-down approach (see Methods and ref. 38). The choice of planar Au electrodes ensures a uniform distribution of the driving electric field in the LiNbO₃ film (see Fig. S3, Supplementary Information), which increases the overlap with the field of the optical mode, also located mostly in the LiNbO₃ volume (see Fig. 2b). The uniform electric field distribution is advantageous for electro-optic (EO) modulation, as effectively leveraging the Pockels effect depends on the overlap between the low-frequency electric field and the optical field within the LiNbO₃ material. The elongated footprint of the metasurface ensures a large active area to fully exploit the high $Q$ of the nonlocal mode, which extends in the direction where the translational symmetry breaking occurs. On the other hand, the metasurface width is reduced to 10 μm to increase the driving field amplitude. Concurrently, the large electrode width minimizes the contact resistance, hence reducing the impact on the device bandwidth. We realized a set of replicas with slight variation of $D_1$ and $D_2$ (see Methods for details and Supplementary Information S3) to target the sample geometry closer to the desired quality factor and resonance position, finally identifying a fabricated replica with $\alpha = 0.19$ and FF = 0.3 as the one matching the original design specifications. Additionally, to demonstrate the generality of this approach, we also investigated a sample with FF = 0.5, which is characterized by a fundamental mode around 1560 nm (see Supplementary Information, sections S3 and S6).

The linear properties and the EO response of the sample are characterized in an epi-reflection configuration by impinging with a linearly polarized, continuous-wave (CW) diode laser tuneable in the optical communication C band (1460 nm–1570 nm) and detecting the reflected signal with a photodiode (details in Methods and Supplementary Information S1). The quasi-BIC resonance wavelength shifts when interrogated away from the Γ point of the photonic band structure, typically by about 10 nm/deg. Therefore, to mitigate the resonance linewidth broadening resulting from the angular spread of the light

from the objective, we use a cylindrical lens to focus the excitation beam onto the objective back-aperture. This approach effectively reduces the numerical aperture, enabling nearly collimated illumination in the direction of the nonlocal mode extension. At the same time, due to the one-dimensional character of the metasurface, we tightly focus the beam on the direction parallel to the nanowires, hence preserving a sizeable beam fluence (see Supplementary Information, Fig. S1a). We determine the resonance position by sweeping the laser wavelength, obtaining the reflectance spectra depicted in Fig. 2f. The sharp quasi-BIC peak is superimposed to a broad Fabry–Pérot fringe caused by the 2 μm-thick SiO₂ layer. A polarization parallel or perpendicular with respect to the LiNbO₃ extraordinary index $n_e$ excites either a transverse-electric (TE) or transverse-magnetic (TM) mode, respectively, resulting in a reflection peak at 1553 nm or 1492 nm, respectively. We also report an experimental redshift of about 5 nm with respect to the COMSOL finite-element simulations of the same structures (see Fig. 2f). This slight discrepancy could be attributed to deviations of the fabricated nanostructure from its nominal design. Due to the asymmetry in the resonance lineshape arising from the interaction between the sharp quasi-BIC and the broad Fabry-Perot fringe, we fitted the experimental data using a Fano profile (see Fig. 2g, h)[39]. We evaluate the resonance quality factor of the TE mode $Q_{exp}$ to be larger than 7500, where $\delta\lambda_0^{exp} \simeq 0.20$ nm is the resonance linewidth retrieved from the fitting. This result is in good agreement with the value calculated from the simulations ($Q_{sim} \simeq 11000$ with $\delta\lambda_0^{sim} = 0.15$ nm). The lower value of $Q_{exp}$ with respect to $Q_{sim}$ might be related to fabrication tolerances and to the difference in excitation between the simulated platform (perfectly planar wavefront) and the real system (partially collimated beam).

We investigate the static EO response of the device by applying a bias $V_{EO} = V_{DC}$ by means of a waveform generator. The shift of the quasi-BIC resonance can be clearly identified in the reflectance spectra of Fig. 3a, superimposed to an unmodulated instrumental artifact. From the central wavelength $\lambda_0$, retrieved by the Fano fit, we estimate an EO induced shift of about $\Delta\lambda_{EO} \simeq 0.05$ nm when $\Delta V_{DC} = 9$ V, close to the value retrieved from the simulated spectra in Fig. 3d ($\Delta\lambda_{EO}^{sim} \simeq 0.06$ nm). We, therefore, estimate a device tuning sensitivity $\Delta\lambda_{EO}/\Delta V_{EO} = 5.6$ pm/V. To observe the dynamic modulation of the reflectance, we apply a sinusoidal driving potential $V_{EO}(t) = \frac{V_{pp}}{2}\sin(2\pi f_{mod}t)$, characterized by a variable peak-to-peak amplitude $V_{pp}$ and modulation frequency $f_{mod}$. We use a lock-in amplifier, referenced by the same waveform generator, to demodulate at $f = f_{mod}$, obtaining the signal oscillation amplitude $P_\omega(f = f_{mod})$. The DC component of the signal $P_\omega(f = 0)$ (namely, the reflectance) is synchronously detected in order to track modulation efficiency variations while changing excitation wavelength. We modulate the signal at $f_{mod} = 100$ kHz, well below the cutoff of our detection system (600 MHz). The relative modulation is calculated by evaluating the ratio $P_\omega(f = f_{mod})/P_\omega(f = 0)$, which corresponds to $\Delta P_\omega/P_\omega = (P_\omega(V = V_{pp}/2) - P_\omega(V = 0))/P_\omega(V = 0)$. The dispersive nature of the reflection spectrum will cause a dependence on $\lambda$ of the modulation amplitude, which is maximized by exciting at the resonance derivative extrema. The maximum modulation amplitude that is obtained from the data in Fig. 3b is $\Delta P_\omega/P_\omega \cong 0.12$ (i.e., 0.24 of peak-to-peak relative modulation), obtained with an input voltage as low as $V_{pp} = 20$ V. This is approximately 5 times smaller than the value obtained from simulations (see Fig. 3e). Such a discrepancy is mostly ascribed to two experimental limitations. On the one hand, the experimental reflectance background is larger than the simulated one (see Fig. 2f), hence reducing the resonance contrast and, consequently, the relative modulation. On the other hand, the presence of fabrication tolerances (e.g., periodicity, asymmetry) and defects in the realized sample, as well as deviations from ideal plane-wave illumination, both contribute to the reduction of $Q_{exp}$. Figure 3c illustrates how electro-optically induced modulation varies with $V_{pp}$. At low modulating amplitudes $\Delta\lambda_{EO} \ll \delta\lambda$ and $\eta$ remains linear with respect to $V_{pp}$. In this

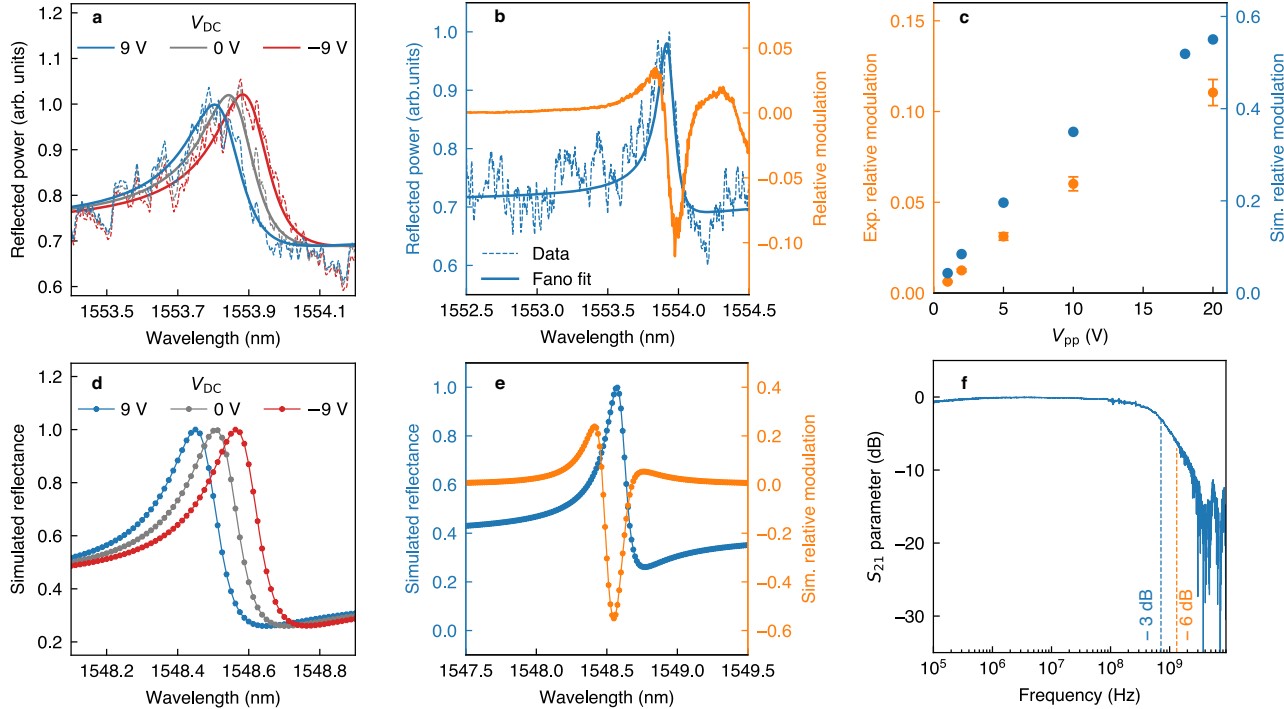

**Fig. 3 | Characterization of the reflected signal modulation by applying DC and AC electrical stimuli. a** Resonance shift upon driving the metasurface with a static bias $V_{DC}$. The dashed lines correspond to the reflected intensity for three different values of $V_{DC}$, the solid lines are obtained by fitting the data with a Fano profile. **b** Relative modulation (in orange), defined as the ratio $P_{\omega}(f = f_{mod})/P_{\omega}(f = 0)$, where $P_{\omega}(f = f_{mod})$ is the signal demodulated at the modulation frequency $f_{mod}$ and $P_{\omega}(f = 0)$ is the DC component of the signal, corresponding to the reflectance. The sinusoidal driving field is characterized by a peak-to-peak amplitude $V_{pp} = 20$ V and a frequency $f_{mod} = 100$ kHz. The bias $V_{DC}$ is set to 0. In blue, the reflectance spectrum (dashed) and its Fano profile fit (solid). The wavelength is swept in steps of 10 pm. **c** Experimental (orange) and simulated (in blue) variation of the maximum

modulation with respect to the peak-to-peak modulating amplitude $V_{pp}$. Experimental points are obtained by modulating with $f_{mod} = 100$ kHz. **d** Reflectance for different applied $V_{DC}$ values and **e** reflectance spectrum and relative modulation (orange dots) simulated using COMSOL Multiphysics. The relative modulation is calculated as $[P_{\omega}(V = V_{pp}/2) - P_{\omega}(V = 0)]/P_{\omega}(V = 0)$, with $V_{pp} = 20$ V. **f** Device frequency response obtained by calculating the parameter $S_{21} = 10\log_{10}[P_{\omega}(f)/P_{\omega}^{max}(f)]$ with a vector-network analyzer (VNA). The VNA oscillator drives the sample with $V_{pp} = 1.5$ V. All the panels refer to theoretical and experimental investigations on the sample with FF = 0.3 and $\alpha = 0.19$. The optical field is polarized along the $z$ axis (see Fig. 2).

regime, the modulation is proportional to the local derivative of the reflectance spectrum, and the linearity of $\Delta n$ with respect to $\mathbf{E}_{EO}$ implies the linearity of $\eta$ versus $V_{pp}$[27]. In this regime, we reach a modulation efficiency−defined as relative modulation per applied Volt − $\eta = \left(\frac{\Delta P_{\omega}}{P_{\omega}}\right)/\left(\frac{V_{pp}}{2}\right)$ of about 0.015 V$^{-1}$ for $V_{pp} < 10$ V. With a larger bias, $\Delta\lambda_{EO}$ is no longer negligible with respect to the resonance width, resulting in a deviation from linearity. We point out that the investigation shows a modulation amplitude of the order of 0.07 for a bias-voltage $V_{pp} = 10$ V. Similar efficiency was reported in the literature only from devices sacrificing the modulation bandwidth to attain larger $\mathbf{E}_{EO}$[28]. We note that the modulation efficiency of the metasurface can be further enhanced via two key strategies: (i) by increasing the quality factor of the quasi-BIC resonance by further reducing the asymmetry factor; and (ii) by suppressing the Fabry-Pérot resonance, which now limits the reflectance contrast of the TM mode, either by removing the silicon substrate or by adjusting the thickness of the SiO$_2$ layer.

We performed the analysis of the modulation efficiency on several fabricated designs, always obtaining values between 0.05 and 0.1 for $V_{pp} = 10$ V (see Supplementary Information S3). To complete the investigation, we studied the efficiency of the TM mode, obtaining a reduced modulation depth (see Fig. S5, Supplementary Information) caused both by the smaller electro-optic coefficient $r_{13} = 10.3$ pm/V[16] and by the slightly larger resonance width of 0.5 nm (Fig. 2h).

Finally, we assessed the modulation speed of the device. To overcome the limitation set by the lock-in amplifier bandwidth (600 MHz), we employed a vector-network analyzer (VNA) operating at frequencies up to 20 GHz and delivering a bias peak-to-peak

amplitude $V_{pp} = 1.5$ V (see Fig. S1b, Supplementary Information). Figure 3f displays the resulting VNA transmission parameter, defined as $S_{21} = 10\log_{10}[P_{\omega}(f)/P_{\omega}^{max}(f)]$, where $P_{\omega}(f)$ is the AC component of the modulated optical signal and $P_s^{max}(f)$ is its maximum value in the considered frequency range. We show a −3 dB (−6 dB) attenuation for a driving frequency $f_{-3dB} \simeq 800$ MHz ($f_{-6dB} \simeq 1.4$ GHz). Possibly, a larger bandwidth could be obtained by reducing the length of the inter-digitated electrodes, which is currently 1.5 mm, since their capacitance scales linearly with the length. The limit for the increased frequency will be imposed by a trade-off between the electrode length and the metasurface area, which needs a minimal size to sustain high $Q$ optical resonances due to the nonlocal character of the mode.

## SHG modulation
The observed modulation shown in the previous section is inherently limited by the imbalance between the high reflected power $P_{\omega}(V = 0)$ and the effective modulated power $\Delta P_{\omega}(\Delta V = V_{pp}/2)$. This imposes a natural upper limit on the modulation amplitude, governed by the intrinsic resonance depth, which corresponds to the ratio between the maximum and the minimum values of the Fano profile over the background. For example, considering the measured reflectance spectrum shown in Fig. 3c, the maximum modulation achievable by a complete shift from the resonance peak $P_{\omega}^{max}$ to off-resonance $P_{\omega}^{off}$ would yield $(P_{\omega}^{max} - P_{\omega}^{off})/P_{\omega}^{max} \simeq 0.3$.

An advantage of modulating resonantly enhanced upconverted signals, such as SHG, is the increased sensitivity due to the quadratic

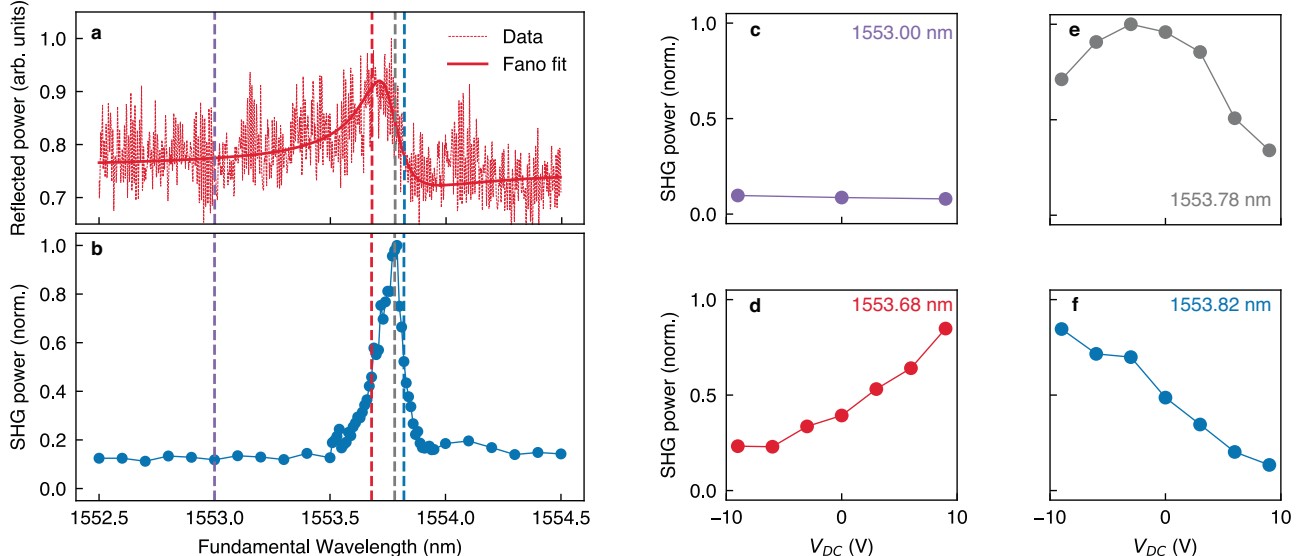

**Fig. 4 | Second harmonic modulation by CD electrical stimuli. a** Spectrum of the reflected signal from the sample with FF = 0.3 and $\alpha$ = 0.19 (see also the Supplementary Information, section S6 for the sample with FF = 0.5). **b** Normalized SHG excitation spectrum, obtained by sweeping the wavelength of the fundamental beam with a constant optical intensity on the sample of 12 kW/cm². **c–f** EO modulation of SHG, excited with four different wavelengths (identified by vertical dashed lines in **a**, **b** depending on the applied DC voltage $V_{DC}$.

dependence on the fundamental electric field, resulting in narrower linewidths. However, a significant drawback is the relatively low absolute conversion efficiency arising from the perturbative nature of the nonlinear process and the small interaction volume inherent to subwavelength devices. To generate substantial SH signals typically requires high peak intensities and, therefore, pulsed lasers. Yet, the broad bandwidths associated with pulsed sources often exceed the resonance wavelength shift $\Delta\lambda_{EO}$ achievable in ultrathin devices, making efficient EO modulation of upconverted signals a challenging task. As a result, very large bias voltages are typically required, which are incompatible with CMOS integration[40]. This limitation can only be circumvented by exploiting laser sources with linewidths significantly narrower than $\Delta\lambda_{EO}$, combined with high-$Q$ resonances. Recently, Anthur et al. demonstrated that the field enhancements associated with the narrow resonances of quasi-BIC modes in a III–V semiconductor metasurface allow to obtain sizeable SHG using continuous-wave (CW) pump sources, effectively compensating the reduced peak intensity[41].

For this reason, we employed the same tuneable CW source used for the linear characterization also as pump for the SHG. We set the laser polarization along the largest component of the $\chi^{(2)}$ tensor $d_{33}$, corresponding to TE illumination. The generated nonlinear signal is then spectrally filtered and detected by means of a dispersive spectrometer equipped with a charge-coupled device (CCD) (see Fig. S1a, Supplementary Information). The spectrum of the collected signal clearly shows the SH peak (see Fig. S6, Supplementary Information) without any incoherent background. We characterize the SH signal as a function of the wavelength of the pump, with a pumping intensity of 12 kW/cm² (see Fig. 4b).

The data reveal a full width at half maximum of the SHG excitation spectrum of $\delta\lambda_{SHG}$ = 0.14 nm $\simeq \frac{\delta\lambda_0^{exp}}{\sqrt{2}}$. The SHG peak aligns with the maximum of the electromagnetic energy stored in LiNbO₃, which notably does not correspond to the maximum of the reflectance spectrum shown in Fig. 4a due to the asymmetry in the Fano line shape (see Fig. S7, Supplementary Information). To tune the SHG emission *via* the Pockels effect, we selected various excitation wavelengths (represented by vertical, dashed lines in Fig. 4b, corresponding to the SHG peak, its inflection points, and to off-resonant excitation), while varying the applied static voltage $V_{DC}$. In case of on-resonance

pumping (as in Fig. 4e), we report a reduction up to a factor of 5 upon application of $\Delta V_{DC} = \pm 9$ V. The change in intensity becomes monotonic when the excitation is tuned on the slopes of the SHG peak (Fig. 4d, f), with $[P_{SHG}(9V) - P_{SHG}(0V)]/P_{SHG}(0V) > 1.1$ of relative SHG intensity variation by applying $\Delta V_{DC} = 9$ V, which is in good agreement with the nonlinear simulation under same excitation conditions (a factor 2 higher, see Fig. S8, Supplementary Information). Notably, we reach a modulation efficiency $\eta = \frac{\Delta P_{SHG}}{P_{SHG}}/\Delta V_{DC} > 0.12\ V^{-1}$ The opposing trends shown in Fig. 4d, f with respect to the sign of $V_{DC}$ confirm the genuine electro-optic origin of the change in the SHG power (namely, $V_{DC} > 0$ V leads to a blue-shift of the resonance, enhancing/suppressing the SHG when excited on the rising/falling slope of the SHG excitation spectrum, respectively). Off-resonant excitation, on the other hand, shows low SHG signal intensity and no electro-optic modulation thereof (Fig. 4c). Importantly, the low SHG signal outside the resonance leads to a high modulation efficiency, close to the simulated value. This is in striking contrast with the linear case, where the modulation is limited by a large reflectance baseline. We note that the maximum modulation that could be obtained with a complete shift of the resonance is only limited by a non-zero off-resonant SHG contribution.

## Discussion

We demonstrate a lithium niobate on Insulator metasurface hosting quasi-BIC resonances for high-speed, efficient EO modulation of optical signals in the C-band employing voltage/bias amplitudes less than 10 V. The same platform can yield sizeable SHG when pumped by CW lasers in the C-band, which can also efficiently be modulated with similar voltages. Our approach achieves 0.07 reflectivity modulation with an oscillating voltage of $V_{pp}$ = 10 V and a modulation efficiency of 0.015 $V^{-1}$, advancing the capabilities of metasurface-based EO modulators, especially in LNOI systems. The interdigitated electrodes design minimizes capacitance, pushing the bandwidth up to 800 MHz and allowing detectable modulation beyond 1 GHz ($f_{-6\ dB}$ = 1.4 GHz). A significant step forward with respect to the previously proposed free-space electro-optic devices is the SHG modulation induced by a static bias. Notably, we report the first experimental demonstration of electrically modulated SHG in subwavelength devices using CW pumping[41]. In particular, we achieved a SHG intensity modulation

exceeding one order of magnitude by applying $\Delta V_{EO} = 9$ V bias (i.e., with efficiency $\eta > 0.12\ V^{-1}$), outperforming the linear counterpart. These results emphasize the value of utilizing narrow linewidth resonances, such as quasi-BIC, in combination with narrowband lasers. These findings demonstrate the feasibility of high-efficiency electro-optic modulation of nonlinear signals driven by low-power CW lasers, establishing a key technological milestone in nanophotonics[41].

## Methods

### Sample fabrication

The sample is fabricated via electron-beam lithography using a hydrogen silsesquioxane (HSQ) resist followed by Argon ion etching via an Inductively Coupled Plasma Reactive Ion-Etching (ICP-RIE) process. The redeposited amorphous $LiNbO_3$ is removed by means of KOH. The tilted sidewalls ($\theta = 65°$, see Fig. 2a, e) resulting from the purely physical etching process have been considered in the structure design. After fixing the filling factor $FF$ and target $\alpha = 0.2$, we account for fabrication imperfections by fabricating several samples with slightly different bar lateral sizes, $D_1$ and $D_2$, in steps of 10 nm, ranging from −20 nm to 20 nm. In such a way we compensate for deviations from the nominal $\alpha$ value. Interdigitated electrodes and interconnects (300 nm Au on 5 nm Cr), obtained via direct laser beam lithography followed by metal evaporation, are placed orthogonally to the nanobars (Fig. 2d).

### Numerical simulations

The metasurfaces linear spectra, the refractive index change induced by Pockels effect, and the second-harmonic generation (SHG) presented in this article have all been simulated by Finite-Element Method (FEM) using COMSOL Multiphysics® 6.2. The translational invariance along $z$ allows us to implement a 2D unit cell (see Figs. 2b and S4, S5 of the Supplementary Information), reducing the computational workload. We set a periodicity $p = 800$ nm for the supercell, which includes two asymmetric nanowires (150 nm high on top of 450 nm $LiNbO_3$ film). Further details on the numerical simulation can be found in Supplementary Information section S2.

### Experimental setup

Supplementary Fig. S1a sketches the experimental setup used for characterizing the electro-optic response of linear and nonlinear signals. A continuous-wave (CW) diode laser source (TOPTICA CTL-1500) provides tuneable excitation in the communication C band (1460 nm–1570 nm), with relative accuracy of 10 p.m. and laser bandwidth <1 kHz. The laser is coupled to free space by means of a polarization-maintaining fiber (Thorlabs, P1-1550PM-FC-5), and its polarization is controlled by a broadband half-wavelength retarder (Thorlabs, AHWP05M-1600). A non-polarizing 50:50 beam splitter separates the excitation from the detection line in an epi-reflection configuration. Dealing with high-$Q$ resonances imposes additional restrictions on the illumination characteristics. The BIC $Q$ factor becomes finite by breaking the translational symmetry along the direction parallel to the 1D crystal axis ($y$ axis in Fig. S2a), as explained in the main text. Using a focusing system produces a superposition of plane waves with $k$ vectors limited by the corresponding numerical aperture, thus smearing the resonance $Q$. This effect is mitigated by a long-focus ($f = 40$ cm) cylindrical lens (Thorlabs, LJ1363RM-C) inserted in the excitation path, which produces an elliptical spot at the back aperture of the objective (Mitutoyo, M Plan Apo NIR HR 50X). Here, the beam is focused along $y$ and collimated along $z$, resulting in a reduced filling of the objective $NA_{y,eff} \simeq 0.1$ NA. This reduces the spread of $k_y$ in the objective front-focal plane, resulting in a beam diameter $D_y \simeq 30\ \mu$m, while keeping a tight focusing in the direction orthogonal to the grating axis ($D_z \simeq 3\ \mu$m). The signal collected by the objective is spectrally separated by means of a dichroic mirror with cut-off wavelength at 950 nm (Thorlabs, DMLP950). The reflected fundamental signal is detected by a fast InGaAs detector (Menlo Systems,

FPD610–FS–NIR). The SHG signal is further spectrally filtered by a combination of filters (long pass at 600 nm and short pass at 900 nm from Thorlabs) to get rid of possible THG and other spurious signals. It is then coupled to a dispersive spectrometer (Andor, Shamrock SR303i) through a multimode fiber and detected by an open-electrode CCD camera (Andor, Newton DU920-OE). The modulating voltage $V(t)$ is provided by an arbitrary waveform generator (GW Instek) and consists in a sinusoidal wave of the form $V(t) = V_{DC} + \frac{V_{pp}}{2}\sin(2\pi f_{mod}t)$. An ultrahigh-frequency lock-in amplifier (Zurich Instrument, UHFLI) is referenced by the same waveform generator and simultaneously detect the AC and DC component of the signal from the photodiode through a split BNC cable.

Supplementary Fig. S1b displays the high-frequency characterization. It has been performed with a vector-network analyser (Keysight, P5004A) driving the sample through an optimized microwave cable (Thorlabs, KMM36). The reflected, modulated optical signal from the sample is coupled into a single-mode fiber and optically amplified by an erbium-doped fiber amplifier (Lumibird CEFA – CHG50) to exploit the full dynamics of a fast, single-mode, unamplified photodetector (New Focus, 45 GHz Photodetector). The electrical signal from the photodetector is then sent to port 2 of the VNA through another microwave cable for the computation of the $S_{21}$ parameter defined in the text.

## Data availability

All data that support the findings of the study are provided in this article and the Supplementary Information file. The raw data generated in this study have been deposited in the Zenodo database under accession code: https://doi.org/10.5281/zenodo.15696618

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

## Acknowledgements

The authors would like to acknowledge financial support by the Swiss National Science Foundation (SNSF) through the program SNF APIC (project id: TMCG-2_213713) and program SNF Synergia (project id: CRSII5_206008).

## Author contributions

A.D.F. performed all the measurements and wrote the first draft of the manuscript. A.S. fabricated the sample. H.W. and J.K. performed the high-speed modulation measurements. E.B. performed the numerical simulations. M.A.V. and L.C. conceived the original idea and revised the manuscript. A.Z. and M.F. supervised the experimental activity and revised the manuscript. M.C. and R.G. supervised the entire project and revised the manuscript.

## Competing interests

The authors declare no competing interests.
