## [Transparent Peer Review file · Nature Communications]

Efficient GHz electro-optical modulation with a nonlocal lithium niobate metasurface in the linear and nonlinear regime

Corresponding Author: Professor Michele Celebrano

Version 0:

Reviewer comments:

Reviewer #1

(Remarks to the Author)

I appreciate the authors' revisions and response to my comments and those of the other reviewers. I think they have overall addressed most of the technical criticisms. The use-cases / applications are still an open question particularly in relation to the extinction ratio, as well as the innovation in relation to prior art (primarily in the context of waveguide modulators); however, I think the results and paper are strong enough to warrant publication in Nature Communications.

Reviewer #2

(Remarks to the Author)

I am writing the following report taking into account the fact that this is essentially the second round of peer review, the submission has been transferred to Nature Communications, and the Editor has accepted the paper's transfer. Overall, the authors' responses to the reviewers' comments were mostly reasonable and appropriate. I evaluate that the preprint of this paper (<https://arxiv.org/abs/2412.03422>) have been cited and made an impact in Dagli et al, <https://arxiv.org/abs/2503.08853> and Soma et al. <https://arxiv.org/abs/2503.17986>. But unfortunately, considering that this paper cites high modulation efficiency and modulation bandwidth as its main achievements, the concern about the poor extinction ratio (ER) that I pointed out in the first review has not been addressed well. As reported in JRD1-based metasurface (Soma et al.), improving the ER is crucial for achieving a clear eye pattern and a low BER. In addition, improvements to the ER are also required in the use for nonlinear layer in metasurface-based optical analog computing. The current manuscript contains some passages that are not appropriate to be published as is in Nat. Commun. .

In conclusion, I just decided to recommend the Editor to publish the paper in Nat. Commun. after the authors have addressed the following minor comments.

1. I do not require manifestation of further improvement in modulation speed, multilevel modulation of PAM-4 or higher, or ER of more than 5 dB in the near future as a condition of publication. However, they should describe a specific approach to improve the metasurface device as it can realize a clear eye pattern in NRZ modulation in the main text, which makes the paper appropriate for publication in Nat. Commun. .
2. The expression "CMOS-compatible voltages/bias" should be deleted. Apparently, "up to 5 V", " $V_{pp} = 10$ V", " $\Delta V_{EO} = 9$ V bias" are not CMOS-compatible. As Soma et al. reported, CMOS-compatible voltage is ~ 1 V or less.
3. In the last sentence of Abstract, they describe "Our findings highlight the transformative potential of LiNbO₃-based metasurfaces for integration into next-generation optical technologies that demand rapid, efficient electrical control of light." They should clarify this in the main text.
4. The expression "groundbreaking", which may look ambiguous and exaggerated from the readers, should be revised. I believe that the authors have learned through a series of peer-review processes that excessive use of exaggerated language such as "groundbreaking" does not lead to high marks from reviewers.

Reviewer #3

(Remarks to the Author)

In their revised manuscript the authors address my raised questions and I still consider the presented work as an important step for achieving optical modulation. I feel that the questions of the other reviewers are also addressed adequately. The topic of the work is suited for Nature Communications and I recommend the publication without further revisions.

We are glad to hear that all reviewers suggest the publication of our manuscript in Nature Communications. While all referees are in favor of publication, one of them is raising few minor comments that deserve to be addressed. Accordingly, please find our detailed response to Referee #2 below.

Replies to Referee #2:

1. I do not require manifestation of further improvement in modulation speed, multilevel modulation of PAM-4 or higher, or ER of more than 5 dB in the near future as a condition of publication. However, they should describe a specific approach to improve the metasurface device as it can realize a clear eye pattern in NRZ modulation in the main text, which makes the paper appropriate for publication in Nat. Commun.

We stress that at this stage we cannot provide characterization of the bit error rate (BER) and eye patterns in Non-Return-to-Zero (NRZ) modulation simply because we are not equipped with the instrumentation to do so. Nevertheless, we agree with the referee that in order to return a clear eye pattern we would need to improve the extinction ratio (ER) of the metasurface. Therefore, we have added the following sentence in the main text to clarify how it would be possible to do so:

“We note that the modulation efficiency of the metasurface can be further enhanced via two key strategies: (i) by increasing the quality factor of the quasi-BIC resonance by further reducing the asymmetry factor; and (ii) by suppressing the Fabry-Pérot resonance, which now limits the reflectance contrast of the TM mode, either by removing the silicon substrate or by adjusting the thickness of the SiO₂ layer”

2. The expression "CMOS-compatible voltages/bias" should be deleted. Apparently, "up to 5 V", " $V_{pp} = 10$ V", " $\Delta V_{EO} = 9$ V bias" are not CMOS-compatible. As Soma et al. reported, CMOS-compatible voltage is ~1 V or less.

We partially agree with the referee. While we stress that CMOS-compatible voltage/bias are typically 3.3V or 5V in digital circuits, for analog operation voltages below 1V may be required. In addition, to achieve high bandwidth operation electro-optical modulation 5V bias are needed. Yet, given that in our paper we report voltage/bias up to 9V for maximum switching, we removed the expression “CMOS-compatible voltages/bias” in the abstract and replace the above expression in the conclusion with a more appropriate “employing voltage/bias amplitudes less than 10V”.

3. In the last sentence of Abstract, they describe "Our findings highlight the transformative potential of LiNbO₃- based metasurfaces for integration into next-generation optical technologies that demand rapid, efficient electrical control of light." They should clarify this in the main text.

We agree that the sentence should be clarified further in the text. Yet, given that it is a rather generic statement and in view of reducing the abstract length (as per request by the editorial office), we decided to drop this sentence.

4. The expression "groundbreaking", which may look ambiguous and exaggerated from the readers, should be revised. I believe that the authors have learned through a series of peer-review processes that excessive use of exaggerated language such as "groundbreaking" does not lead to high marks from reviewers.

We agree with the referee on the fact that these terms must be limited to avoid overselling the outcomes of the paper. Therefore, we have removed the term "groundbreaking". In particular, we have rephrased the sentence:

"In this study, we introduce and demonstrate a groundbreaking use of nonlocal metasurfaces based on monolithic lithium niobate on insulator (LNOI) platform. By leveraging a high-quality factor resonance, we achieve efficient intensity modulation of free-space signals within the telecom C band."

With the following:

"In this study, we leverage high-quality factor resonances in nonlocal metasurfaces based on monolithic lithium niobate on insulator (LNOI) platform to realize efficient intensity modulation of free-space signals within the telecom C band."